# Third harmonic characterization of antiferromagnetic heterostructures

Yang Cheng[1,2], Egecan Cogulu[3], Rachel D. Resnick[1], Justin J. Michel[1], Nahuel N. Statuto[3], Andrew D. Kent[3] & Fengyuan Yang [1✉]

Electrical switching of antiferromagnets is an exciting recent development in spintronics, which promises active antiferromagnetic devices with high speed and low energy cost. In this emerging field, there is an active debate about the mechanisms of current-driven switching of antiferromagnets. For heavy-metal/ferromagnet systems, harmonic characterization is a powerful tool to quantify current-induced spin-orbit torques and spin Seebeck effect and elucidate current-induced switching. However, harmonic measurement of spin-orbit torques has never been verified in antiferromagnetic heterostructures. Here, we report harmonic measurements in Pt/$\alpha$-Fe$_2$O$_3$ bilayers, which are explained by our modeling of higher-order harmonic voltages. As compared with ferromagnetic heterostructures where all current-induced effects appear in the second harmonic signals, the damping-like torque and thermally-induced magnetoelastic effect contributions in Pt/$\alpha$-Fe$_2$O$_3$ emerge in the third harmonic voltage. Our results provide a new path to probe the current-induced magnetization dynamics in antiferromagnets, promoting the application of antiferromagnetic spintronic devices.

[1] Department of Physics, The Ohio State University, Columbus, OH 43210, USA. [2] Department of Electrical and Computer Engineering, and Department of Physics and Astronomy, University of California, Los Angeles, CA 90095, USA. [3] Department of Physics, Center for Quantum Phenomena, New York University, New York, NY 10003, USA. ✉email: yang.1006@osu.edu

Antiferromagnetic (AFM) spintronics is an emerging research field with great potential for ultrafast, energy-efficient future technology[1–7]. In the past several years, current-induced switching of AFM Néel order has been demonstrated in several antiferromagnetic materials, including metallic AFM CuMnAs and $Mn_2Au$ as well as heavy-metal (HM)/AFM-insulator bilayers such as Pt/NiO and Pt/$\alpha$-$Fe_2O_3$[8–15]. These recent developments generate intense interests in active AFM devices. However, there is ongoing debate on the mechanism of the Néel order switching, which could be induced by spin-orbit torque (SOT) or the magnetoelastic effect as well as artifact signals from heavy metals and the relation to AFM grain morphology[8,9,13,16,17].

Lock-in detection technique has been widely used to investigate current-induced spin torque contributions in HM/ferromagnetic (FM) systems by measuring the first and second harmonic voltages[18–20]. For AFMs, the second harmonic measurement has been used for identifying 180° Néel vector reversals in CuMnAs[21]. However, it requires that the AFM has both broken time and space inversion symmetry. Whether harmonic measurement can be used in characterizing the current induced effect in other AFMs is still an open question[22–26].

In this article, we report harmonic measurements in HM/AFM bilayer Pt/$\alpha$-$Fe_2O_3$. As compared to the HM/FM bilayers where spin torques only contribute to the second harmonic signals, our results shown that for HM/AFMs, the damping-like SOT, as well as the magnetoelastic effect, appear in the third harmonic response. Our theoretical modeling, together with the temperature-dependent harmonic measurements, indicate that the magnetoelastic effect could have an important contribution to current-induced AFM switching.

## Results

$\alpha$-$Fe_2O_3$ is an easy plane AFM at room temperature with the Néel order in *ab*-plane (0001). Due to the Dzyaloshinskii–Moriya interaction (DMI), there is a small in-plane canting of Néel order, which exhibits a very weak moment[27]. We grow epitaxial $\alpha$-$Fe_2O_3$ films on $Al_2O_3$ (0001) substrate by off-axis sputtering[8,28,29]. X-ray diffraction scan (see Supplemental Materials) of a 30 nm $\alpha$-$Fe_2O_3$ film shows Laue oscillations, demonstrating high crystal quality of the $\alpha$-$Fe_2O_3$ film. Subsequently, we grow a 5 nm Pt layer on $\alpha$-$Fe_2O_3$ by off-axis sputtering at room temperature. We pattern the Pt/$\alpha$-$Fe_2O_3$ bilayers into a 5 µm wide Hall cross using photolithograph and Ar ion etching, as schematically shown in Fig. 1a. For the harmonic measurement, we apply a 4 mA ac current **I** at 17 Hz and measure the first ($1\omega$), second ($2\omega$), and third ($3\omega$) harmonic voltages using a lock-in amplifier.

**First harmonic Hall signals**. We first show the angular dependence of first harmonic voltage for a Pt(5 nm)/$\alpha$-$Fe_2O_3$(30 nm) bilayer at a temperature ($T$) of 300 K in the presence of an in-plane magnetic field (**H**) from 0.3 to 14 T. Figure 1b schematically illustrates the two spin sublattices $\mathbf{m}_{A(B)}$ of $\alpha$-$Fe_2O_3$ with the in-plane magnetic field applied at an angle $\varphi_H$ relative to the $\mathbf{x}$ axis. We also define the unit vector of Néel order $\mathbf{n} = \frac{\mathbf{m}_A - \mathbf{m}_B}{|\mathbf{m}_A - \mathbf{m}_B|}$ and net magnetization $\mathbf{m} = \mathbf{m}_A + \mathbf{m}_B$, as shown in Fig. 1c. The orientations of these relevant vectors, $\mathbf{m}_A$, $\mathbf{m}_B$, $\mathbf{n}$, $\mathbf{m}$, and $\mathbf{H}$ are represented by their polar angle $\theta$ and azimuthal angle $\varphi$. Figure 1d shows the $\varphi_H$-dependence of first harmonic voltage $V_{1\omega}$ which is the same as the transverse spin Hall magnetoresistance (TSMR) in DC measurements (see Eq. S1 in Supplementary Materials for more details). Based on the theory of spin Hall magnetoresistance (SMR), when the current is applied along the $\mathbf{x}$ direction, the generated spin current with spin polarization $\boldsymbol{\sigma}$ is along the $\mathbf{y}$ direction. Depending on the relative angle between $\boldsymbol{\sigma}$ and $\mathbf{n}$, the transverse voltage $V_{1\omega} \propto n_x n_y$[30,31]. For our $\alpha$-$Fe_2O_3$ films, we

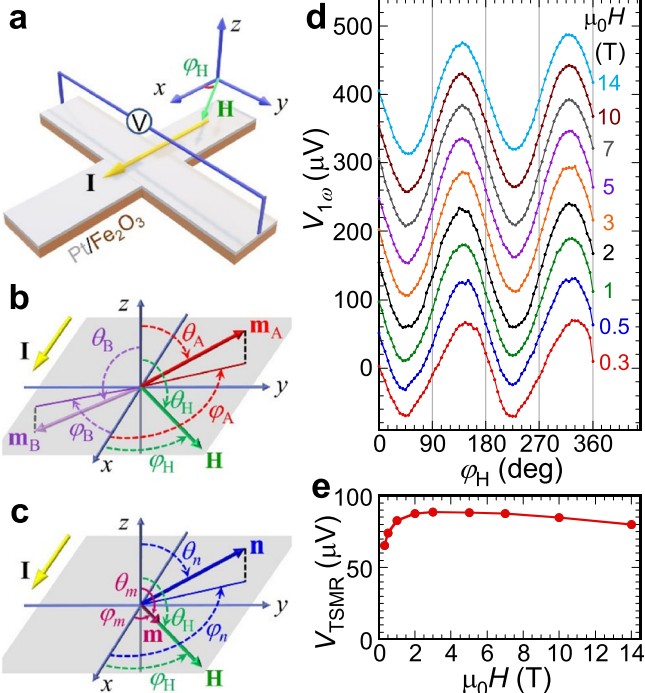

**Fig. 1 Experimental geometry and first harmonic results.** Schematics of **a** a Pt/$\alpha$-$Fe_2O_3$ Hall cross with a 5 µm channel width, **b** two spin sublattices $\mathbf{m}_{A(B)}$, and **c** unit vector of Néel order $\mathbf{n}$ and net magnetization $\mathbf{m}$ of $\alpha$-$Fe_2O_3$ in the presence of an in-plane magnetic field $\mathbf{H}$ within a spherical coordinate system with polar angle $\theta$ and azimuthal angle $\varphi$ for each of the vectors: $\mathbf{m}_A$ (red), $\mathbf{m}_B$ (purple), $\mathbf{n}$ (blue), $\mathbf{m}$ (magenta), and $\mathbf{H}$ (green). **d** In plane angular dependence of first harmonic Hall voltage $V_{1\omega}$ for a Pt(5 nm)/$\alpha$-$Fe_2O_3$(30 nm) bilayer at different magnetic fields from 0.3 to 14 T at 300 K. **e** Field dependence of transverse spin Hall magnetoresistance voltage $V_{TSMR}$ extracted from the fitting in **d** by Eq. (1).

showed previously[8,29] that the spin-flop transition occurs at the critical field of <1 T, where the Néel order is perpendicular to the magnetic field, $\mathbf{n} \mathbf{H}$. Then,

$$V_{1\omega} = -V_{TSMR} \sin 2\varphi_H. \quad (1)$$

Such TSMR has been demonstrated in many Pt/AFM bilayer systems[31–33]. Fitting the angular-dependent $V_{1\omega}$ curves in Fig. 1d with Eq. (1), we extract $V_{TSMR}$ for each value of the magnetic field, which is plotted in Fig. 1e. The magnitude of $V_{TSMR}$ saturates near $\mu_0 H = 1$ T, which is consistent with our previous results[8], indicating single domain AFM state at $\mu_0 H > 1$ T. One notes that there is a small decrease of $V_{TSMR}$ at high field. This is due to the tilting of the AFM spins at high field, which lowers the value of Néel vector $\mathbf{n}$[34].

**Second harmonic Hall signals**. In addition to the first harmonic signals, we simultaneously measure the second and third harmonic voltages. For the second harmonic voltage $V_{2\omega}$, our modeling (see Supplementary Materials for details) shows that it consists of two components, the field-like (FL) SOT and the spin Seebeck effect (SSE), which can be written as,

$$V_{2\omega} = V_{2\omega}^{FL} + V_{2\omega}^{SSE} = V_{TSMR} \frac{H_{FL}}{H} \cos(2\varphi_H)\cos\varphi_H + V_{SSE}\cos\varphi_H, \quad (2)$$

where $H_{FL}$ is the effective field of field-like torque and $V_{SSE}$ is the SSE voltage. Figure 2a shows the in-plane angular dependent $V_{2\omega}$ curves at different magnetic fields from 1 to 14 T. Each curve in Fig. 2a is fitted by Eq. (2), such as those shown in Fig. 2b for $\mu_0 H = 5$ T.

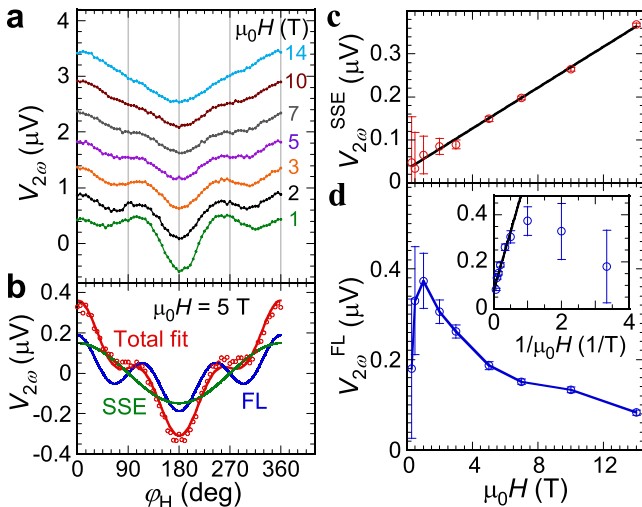

**Fig. 2 Second harmonic results. a** In-plane angular dependence of second harmonic Hall voltage $V_{2\omega}$ at different magnetic fields for a Pt(5 nm)/α-Fe$_2$O$_3$(30 nm) bilayer at 300 K. **b** Angular dependence of $V_{2\omega}$ at, 5 T where the blue and green curves are contributions from the field-like torque and spin Seebeck effect, respectively, while the red curve is the total fit by Eq. (2). Field dependence of **c** spin-Seebeck effect (SSE) contribution $V_{2\omega}^{SSE}$ and **d** field-like (FL) torque contribution $V_{2\omega}^{FL}$ where the inset of **d** shows the corresponding $1/H$ plots and linear fitting. $V_{2\omega}^{SSE}$ exhibits a linear dependence of field. $V_{2\omega}^{FL}$ shows a $1/H$ dependence at $\mu_0 H > 1$ T. Error bars represent fitting uncertainty.

We extract the magnitudes of these two contributions at different magnetic fields as shown in Fig. 2c, d. Instantly, we can find the differences between AFMs and their FM counterparts. For FMs, the SSE saturates when the total magnetization is aligned with the external magnetic field, while the SSE in Pt/α-Fe$_2$O$_3$ linearly increases with $H$ as shown in Fig. 2c because when $H$ exceeds the spin-flop field, the net magnetization in α-Fe$_2$O$_3$ is $\mathbf{m} = \chi_\perp \mathbf{H}$, resulting in $V_{SSE} \propto \mathbf{m} \propto \mathbf{H}$. This is also consistent with previous work on Pt/Cr$_2$O$_3$ bilayers where the SSE is observed when Cr$_2$O$_3$ is in the spin-flop state[35]. The SSE in AFMs originates from the tilting-induced net magnetic moment which is parallel to the external field.

For the field-like torque term shown in Fig. 2d, $V_{2\omega}^{FL}$ first increases with field at $\mu_0 H < 1$ T and then decreases at higher fields. The inset of Fig. 2d gives the $1/\mu_0 H$ dependence of $V_{2\omega}^{FL}$, which clearly shows that the $1/H$ dependence as predicted by Eq. (2) is valid at high fields. This is because the precondition of Eq. (2) is the single domain state of α-Fe$_2$O$_3$, which is fulfilled only when $\mu_0 H > 1$ T, as demonstrated by the first harmonic data shown in Fig. 1e. From the fitting, we obtain $H_{FL} = 35$ Oe, which is consistent with previous reports, while the Oersted field contribution in our Hall cross is only ~5 Oe[9].

**Third harmonic Hall signals**. In our modeling of the harmonic signals for Pt/α-Fe$_2$O$_3$, a striking difference as compared with FM systems is that the damping-like (DL) torque contribution does not appear in the second harmonic when the field is rotated in the x-y plane[23,36], but in the third harmonic voltage. A detailed study of the third harmonic voltage (See Supplementary Materials) reveals that there are three terms in $V_{3\omega}$,

$$V_{3\omega} = V_{3\omega}^{DL} + V_{3\omega}^{ME} + V_{3\omega}^{\Delta R}$$

$$= V_{TSMR}\left(-\frac{H_{ex}H_{DL}^2}{4H(H+H_{DM})\left(H_K+H_{DM}\left(\frac{H+H_{DM}}{2H_{ex}}\right)\right)} + \frac{H_{ex}H_{ME}}{4H(H+H_{DM})}\right)\sin 4\varphi_H$$

$$+ \frac{1}{8}\Delta V_{TSMR}\sin 2\varphi_H,$$

$$(3)$$

where $V_{3\omega}^{DL}$, $V_{3\omega}^{ME}$, and $V_{3\omega}^{\Delta R}$ are the damping-like torque, magnetoelastic (ME) effect, and change of the resistivity (ΔR) term, respectively. $H_{ex}$, $H_{DM}$, $H_K$, $H_{DL}$, and $H_{ME}$ are the exchange field, DMI effective field, easy-plane anisotropy field, damping-like torque effective field, and ME-induced effective easy-axis anisotropic field along $\mathbf{x}$, respectively. $V_{3\omega}^{\Delta R}$ mainly originates from the change of Pt resistivity due to the applied current. In previous reports of electrical switching of AFMs, thermally-induced Pt resistivity change has led to saw-tooth shaped artifact in switching signals[8–10,16,37]. And there could be a very minor contribution to $V_{3\omega}^{\Delta R}$ due to the heating induced soften of magnetization given the very high Néel temperature of α-Fe$_2$O$_3$[20]. Equation (3) reveals why damping-like torque and ME only appear in the third harmonic voltage as $H_{DL}^2$ and $H_{ME} \propto I^2$[9], whereas in FMs, linear dependence on $H_{DL}$ appears in the second harmonic voltage.

Figure 3a shows the in-plane angular dependence of $V_{3\omega}$ at different magnetic fields, which is fitted by Eq. (3). Figure 3b, c shows the fitting of $V_{3\omega}$ for 0.3 and 10 T, respectively, with separate $\sin 2\varphi_H$ and $\sin 4\varphi_H$ components. At 0.3 T, the $V_{3\omega}^{DL}$ and $V_{3\omega}^{ME}$ contribution with a $\sin 4\varphi_H$ dependence is comparable to the $V_{3\omega}^{\Delta R}$ term with a $\sin 2\varphi_H$ dependence. However, at 10 T, $V_{3\omega}^{\Delta R}$ dominates the third harmonic voltage. Figure 3d shows $V_{3\omega}^{\Delta R}$ as a function of the magnetic field and Fig. 3e shows $V_{3\omega}^{\Delta R}$ normalized by $V_{TSMR}$, which is essentially field independent, indicating its nonmagnetic origin. Since $V_{3\omega}^{DL}$ and $V_{3\omega}^{ME}$ have the same angular dependence, Fig. 3f combines them as $V_{3\omega}^{DL+ME}$, which shows a quick decay as the field increases.

To better understand the contribution from $V_{3\omega}^{DL}$ and $V_{3\omega}^{ME}$, we make the same harmonic measurement at lower temperatures. For bulk α-Fe$_2$O$_3$, when the temperature is lower than the Morin transition temperature $T_M \sim 260$ K, it experiences a spin reorientation transition, where the α-Fe$_2$O$_3$ becomes an easy-axis AFM[38]. However, for (0001)-orientated α-Fe$_2$O$_3$ thin films, $T_M$ is much lower or even does not exist due to epitaxial strain[8,39,40] as confirmed by the similar angular dependence in the DC[29] and harmonic measurements. Thus, in our measured temperature range (100–300 K) the α-Fe$_2$O$_3$ is still an easy-plane AFM. Figure 4a shows the normalized $V_{3\omega}^{DL+ME}$ by $V_{TSMR}$ at $T = 300, 200$, and 100 K, which is fitted by Eq. (3). We find that $V_{3\omega}^{DL+ME}$ decreases at lower temperatures and basically vanishes at 100 K. The effective anisotropic field of the magnetoelastic effect $H_{ME}$ is induced by thermoelastic stress $\Delta\sigma$[41]. We use the finite-element simulation (see Supplementary Materials for more details) to estimate $\Delta\sigma$ in our Hall cross at the corresponding temperatures. Then we obtain $H_{ME} = \frac{2\lambda_s\Delta\sigma}{M_0}$[9,42], where $\lambda_s = 1.4 \times 10^{-6}$ is the magnetostrictive coefficient of α-Fe$_2$O$_3$ and $M_0 = 759$ emu/cm$^3$ is the sublattice magnetization[43].

Figure 4b shows the simulated $H_{ME}$ together with the fitted $H_{ME} - H_{DL}^{eff}$, where $H_{DL}^{eff} = \frac{H_{DL}^2}{H_K + H_{DM}\left(\frac{H+H_{DM}}{2H_{ex}}\right)}$, from Eq. (3) at different temperatures using $H_{ex} = 9 \times 10^6$ Oe and $H_{DM} = 1.78 \times 10^4$ Oe[44,45]. From Fig. 4b, we can estimate the magnitude of $H_{ME}$ in our experiment is ~0.1 Oe at 300 K. The damping-like torque effective field, however, is challenging to quantify here since it has a quadradic dependence. In Fig. 4b, the simulated $H_{ME}$ is slightly larger than the values extracted from the experimental data and the difference is larger at higher temperatures. This could be due to the parameter choice or the contribution of $H_{DL}^{eff}$. If we believe the larger $H_{ME}$ is due to $H_{DL}^{eff}$, and assume the easy-plane anisotropic field $H_K$ ~100 Oe[46], we can evaluate that $H_{DL}$ has the order of 1 Oe. One notes that this is an order of magnitude smaller than $H_{FL}$, which may be related to the insulating nature of α-Fe$_2$O$_3$. It is known that FL(DL)-SOT is determined by the imaginary (real) part of spin mixing

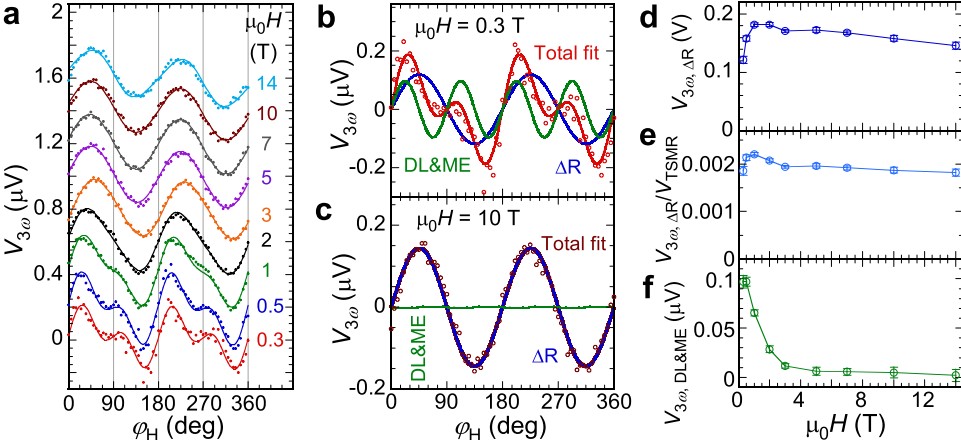

**Fig. 3 Third harmonic results. a** In-plane angular dependence of third harmonic Hall voltage $V_{3\omega}$ at different magnetic fields for a Pt(5 nm)/$\alpha$-Fe$_2$O$_3$(30 nm) bilayer at 300 K. Angular dependence of $V_{3\omega}$ at **b** 0.3 T and **c** 10 T, where the blue curve is from the change of Pt resistivity ($\Delta R$), the green curve is from the damping-like (DL) torque and the magnetoelastic effect (ME) (they have the same angular dependence), and the red curve is the total fit by Eq. (3). Field dependencies of **d** $V_{3\omega}^{\Delta R}$, **e** $V_{3\omega}^{\Delta R}$ normalized by the transverse spin Hall magnetoresistance (TSMR) signal $V_{TSMR}$, and **f** $V_{3\omega}^{DL+ME}$. Error bars represent fitting uncertainty.

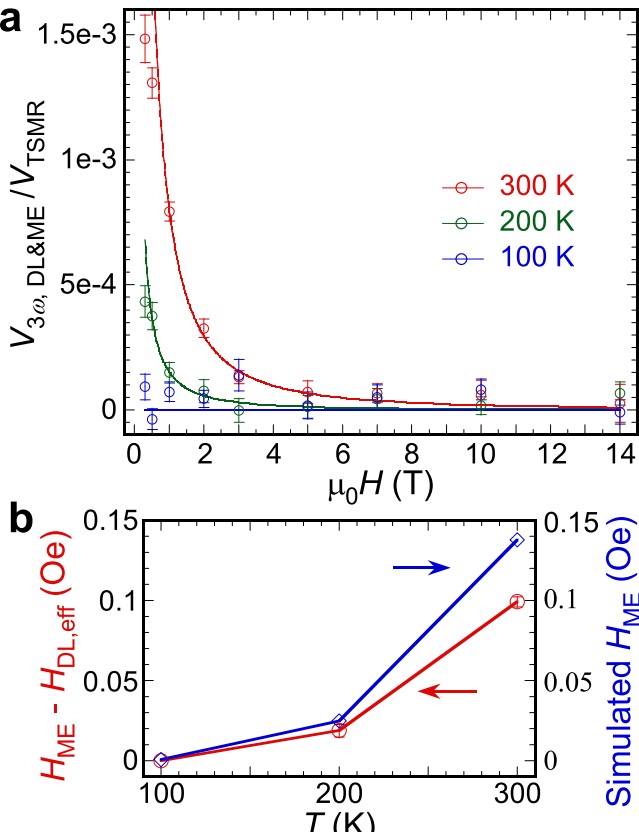

**Fig. 4 Third harmonic components. a** Damping-like (DL) torque and magnetoelastic effect (ME) contribution $V_{3\omega}^{DL+ME}$ normalized by the transverse spin Hall magnetoresistance (TSMR) signal $V_{TSMR}$ as a function of applied magnetic field at 300, 200, and 100 K. Error bars represent fitting uncertainty. **b** Temperature dependence of $H_{ME} - H_{DL}^{eff}$ (red), where $H_{DL}^{eff} = \frac{H_{DL}^2}{H_K + H_{DM}(\frac{H+H_{DM}}{2H_{ex}})}$, extracted from the fitting in **a** by Eq. (3) and simulated $H_{ME}$ (blue) from the magnetic anisotropy energy due to magnetoelastic effect.

conductance. In HM/ferromagnetic-insulator heterostructures such as Pt/Y$_3$Fe$_5$O$_{12}$ and Pt/EuS, the imaginary part of spin mixing conductance is an order of magnitude larger than its real part (see Supplementary Materials for more discussion)[47–49].

Further research in HM/AFM-insulator is needed to better understand the SOTs in AFM heterostructures.

## Discussion

As harmonic measurements have been used in many FM materials, we show that they also serve as a powerful tool in investigating current-induced effects in HM/AFM systems. Usually, AFMs have very large magnetic anisotropies and remain in multiple-domain states even under a strong magnetic field. The multiple-domain state of AFMs hinders the quantitative analysis of current-induced magnetization change. In this regard, $\alpha$-Fe$_2$O$_3$ is different from other AFMs and reaches single-domain state at a relatively low field, making it an ideal platform for harmonic characterization. Our modeling results match well with the experimental data, indicating the validity of our model.

From the harmonic measurement, we find that $V_{2\omega}^{FL}$ and $V_{2\omega}^{SSE}$ have similar in-plane angular dependence as those in FMs because the current-induced FL torque and SSE act similarly on AFMs as on FMs. The third harmonic voltage shows the key difference between AFMs and FMs where both DL torque and ME terms play an important role for AFMs. The quadratic dependence of $H_{DL}$ is not surprising, as previous theoretical and experimental works have confirmed that reversing the current direction by 180° does not affect the switching of Néel order by damping-like torque[15]. The magnitude of $H_{ME}$ is estimated to be ~0.1 Oe at a current density of $J = 1.6 \times 10^{11}$ A/m$^2$. Although we cannot precisely obtain the magnitude of $H_{DL}$, based on previous harmonic measurements in FMs, $H_{DL}$ is 11.7 Oe at $J = 1.0 \times 10^{11}$ A/m$^2$ for Pt/Co and 12.3 Oe at $J = 2.1 \times 10^{11}$ A/m$^2$ for Pt/TmIG[18,19]. Our spin Hall magnetoresistance measurement in Pt/$\alpha$-Fe$_2$O$_3$ reveals a large spin mixing conductance $G_{\uparrow\downarrow} = 5.5 \times 10^{15}$ $\Omega^{-1}$m$^{-2}$[29], comparable to the best Pt/FM interfaces[50,51]. Thus, combined with our previous evaluation, we expect that $H_{DL}$ is of the order 1 Oe under our experimental conditions, which is one to two orders of magnitude larger than $H_{ME}$. However, since $H_{ME} \propto I^2$, $H_{ME}$ can reach ~1 Oe under the current density for switching measurement. Considering the relatively small easy-axis anisotropic field in $\alpha$-Fe$_2$O$_3$, ME may offer an important contribution to help overcome the energy barrier for AFM switching.

## Methods

**Sample preparation.** Epitaxial $\alpha$-Fe$_2$O$_3$ films are grown on Al$_2$O$_3$(0001) substrates using radio-frequency off-axis sputtering in a 12.5 mTorr sputtering gas of Ar + 5% O$_2$ at a substrate temperature of 500 °C. Pt/$\alpha$-Fe$_2$O$_3$ bilayers, as well as Pt single

layers on $Al_2O_3$, are patterned into the Hall cross structure using photolithography and Argon ion milling for electrical measurements.

**Harmonic measurement.** The in-plane angular dependence measurements are performed using a Quantum Design 14 T Physical Property Measurement System (PPMS). An ac current *I* with an amplitude of 4 mA and frequency 17 Hz is applied by a Keithley 6221 current source while the harmonic voltage is measured by Stanford SR865A lock-in amplifier.

## Data availability

All data generated in this study are presented in the paper and the Supplementary Information.

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

## Acknowledgements

This work was primarily supported by the Department of Energy (DOE), Office of Science, Basic Energy Sciences, under Grant No. DE-SC0001304 (film growth, harmonic measurements, and analysis), and partially supported by the Air Force Office of Scientific Research under grant FA9550-19-1-0307 (sample patterning and X-ray diffraction).

## Author contributions

Y.C., R.D.R., and J.J.M. fabricated the samples. Y.C. performed the harmonic measurements, analyzed the data, built the theoretical model, and drafted the manuscript. E.C. and N.N.S. contributed to the second harmonic experiment. F.Y. and A.D.K. supervised the project. All authors discussed the results and commented on the manuscript.

## Competing interests

The authors declare no competing interests.

**Additional information**

