## [Peer Review File · Nature Communications]

Reviewers' Comments:

Reviewer #1:

Remarks to the Author:

I have carefully read the response letter, the modified manuscript, and the SI. While I see some improvements, my main concerns remain unresolved or vaguely addressed. I still find the manuscript interesting but identify several shortcomings in its current state, which I will delineate through my replies to the authors' comments. In the present form, I cannot recommend this work for publication in Nature Communications.

Re: Response2 – Here, I meant S1 through S6 including the sub-numberings (S1-1, S1-2, S2-1, etc.). I have given another try at understanding the steps during this second revision and I still struggle to make a physical meaning out of the mathematical description in Eq. S6-2. The strength of harmonic SOT analysis lies in its general applicability to other similar systems of interest. This is best ensured if the developed method passes through the various sanity checks. I am curious if the authors have tried to set H_{DM} to zero and performed the simulations. Because in the absence of H_{DM} and H (or very small H), the system can be treated like two ferromagnets with opposite magnetization directions as far as H_{DL} is concerned. In this case, I strongly doubt that an in-plane tilting of the magnetization would occur because the opposite sublattices will act symmetrically to H_{DL} , which will tilt them out-of-plane in opposite directions but I do not see why an in-plane tilting would occur due to the antiferromagnetic coupling between the sublattices, unless there is a phenomenon that breaks this symmetry. When I set H_{DM} and H to (nearly) zero in Eq. S6-2 $\delta\text{-}\Phi$ term diverges to infinity, which is unphysical.

Another sanity check is concerning the simulations. The authors have used a single set of parameters and performed the simulations for a fixed H_{DL} direction. What if you set $H_{DL} = -1$ Oe? Do you still get the same sign of $\delta\text{-}\Phi$ vs. Φ_H or the opposite sign? The same (opposite) sign would mean that this signal may show up in the third (second) harmonic. What about setting H_{DM} to zero? H_{DM} is specific to this material but it is not a general attribute to antiferromagnets. If the third harmonic response is somewhat linked to the presence of the DMI, some of the paper's claims (developing a general method to characterize SOTs) should be revisited.

Nevertheless, unless significantly improved, even though fully clarified from the above ambiguities, I find the method presented here highly inaccessible to an average researcher who might be dealing with SOTs in antiferromagnets, let alone the more general audience.

Re: Response3: This response, unfortunately, does not address my main concern that "the current method will fail to identify the torque direction". They use a prototype SOT material Pt from which we know the vector form of the SOTs and their approximate amplitudes. If the DL-SOT direction were to be characterized in a new material, the presented method would fail to do so since there is no vector information on the third harmonic signal. Not to mention the added complexity of having an accurate estimation of the ME contributions..

Re: Response4: OK, once the appearance of DL-SOT contribution to the third harmonic signal is clarified.

Re: Response5: I see and respect that the authors mainly rely on literature values of similar materials or exceptionally large values reported in Pd/Co multilayers to justify the large FL-SOT. First of all, the early findings in Pd/Co multilayers have been generally associated with spurious signals as recent work has failed to reproduce them (See: Phys. Rev. Applied 7, 014004 2017).

The only original argument I can identify in the authors' response is that the antiferromagnets have distinct properties than the ferromagnets. This is surely the case, but without further intuitive explanation, this argument remains ambiguous and not satisfactory when explaining an important finding that the FL-SOT is ~ 35 times larger than the DL-SOT. With the current understanding of the spin-torque phenomenon, this is unphysical, or a fundamentally new understanding is at stake.

Re: Response6: I would be reluctant to accept that "this work shows that harmonic measurements can be smoothly generalized in HM/AFM systems and provides a powerful tool to the community to analyze all current-induced effects". A significant improvement is required to reach this point. Performing measurements in different antiferromagnetic systems would be cumbersome at this stage. A suggestion is to perform simulations spanning a parameter space that would also be of interest to other common antiferromagnetic systems (NiO, Cr2O3, CoO, etc.) and showing/discussing them in the main text. This would be useful guidance for future similar studies.

Additional comments:

-There is an inconsistency in the manuscript regarding the use of data at low and high fields to characterize the DL-SOT and FL-SOT. The authors rely on the high field values for the characterization of the FL-SOT due to possible signal contamination at low fields from the multidomain state. However, the $\sin(4\phi)$ signal in the third harmonic channel, presumably responsible for the DL-SOT, only appears at and below 1T with a magnitude that could be unequivocally associated with a signal having a physical origin. Above this field value, the signal is a carbon copy of the first harmonic signal, which is understood as the resistivity term (not of SOT origin). It is not possible to use harmonic analysis for materials in a multidomain state because of the unavoidable current-induced domain wall motion generated harmonic signals that would overlap the actual signal. The fact that the DL+ME terms are absent in the single domain state casts further doubt about the real origin of the third harmonic signal in this sample.

-In response to Reviewer1 the authors perform current-dependence of the 2nd and 3rd harmonic measurements and show them in the SI. It seems like at 1 mA no signal is present at all and at higher currents, the expected dependencies are not always strictly observed. The authors should comment on these inconsistencies.

Reviewer #2:

Remarks to the Author:

I thank the authors for considering my comments/criticisms in their modified draft.

1) However, the answer of the authors regarding my main critique 1 is unfortunately not satisfying and my doubt about the symmetry of the second harmonic component of the transverse voltage signal remains.

As the authors correctly note: for strong applied magnetic fields above the spin-flop transition where the Neel vector aligns perpendicular to the magnetic field for all field angles, the second harmonic signal generated by the field-like torque follows a $\cos(\phi_H)$ dependence in their model and therefore is "odd under reversal".

However, in my opinion, the symmetry of the second harmonics should also follow an "odd under field reversal" angular dependence at small applied field strengths, even if axial and/or easy-plane anisotropies prevent the full Neel vector from being oriented perpendicular to the applied field. Thus, I still cannot understand how a $\cos(2\phi_H)$ can appear as the dominant component of the measured second-harmonic signal even in a fictitious multidomain state at low fields, since all contributions from all the individual domains should have no or an "odd" field dependence due to the field induced tilt of the Néel vector. Since the origin of this largest of all higher harmonic signal is not traced, doubts arise also about the interpretation of the much smaller third harmonic signal. The authors should therefore replace the vague statement that the angular dependence is "complicated and uncertain" by a comprehensible interpretation of the even under field reversal contribution of the second harmonic signal.

2) Current-amplitude dependence of the signal of the 2nd second harmonic:

◇ The authors now show in Fig. S7(a) the current dependence of the signal contribution attributed to the spin Seebeck effect, i.e., $\sim \cos(\phi_H)$. Surprisingly, the current dependence seems to follow a linear rather than the expected quadratic dependence.

Certainly very illuminating with respect to the interpretation of the harmonic signals is the current dependence of the 2nd harmonic contributions attributed to the field-like torque, which include both the "odd" term $\sim \cos(2\phi_H) \cos(\phi_H)$ and the questionable "even" term $\sim \sin^2(\phi_H)$. I would be very interested to see the current dependence of these two contributions both at low field below the spin flop transition and at high field above the transition.

3) I thank the authors for explaining to me in detail how H_{FL} , H_{DL} , and H_{ME} were determined.

4) As a minor point: to achieve current-induced switching of the AF order and detection of the 2nd harmonic of the 180 degree inversion, the requirement of broken space inversion symmetry is also realized at an AF/HM interface and bulk space inversion symmetry of the AF is not necessarily required I think.

5) and 6): I agree with the authors, thank you.

In summary, I think that the problem with the symmetry of the 2nd harmonic signals, which is important, has not yet been solved in the current draft of the paper. Perhaps the analysis with respect to the current dependences will help identifying the origins of the different 2nd harmonic terms.

Reviewer #3:

Remarks to the Author:

I was a reviewer of this paper at the previous round (before it was transferred to Nature Communications). I have read the authors' response to all the comments (mine as well as other Reviewers) and find them satisfactory. Moreover, after reading the comments of my colleagues I agree that the paper is more suitable for Nature Communications and would recommend to publish it in the present form.

Response to reviews on manuscript NCOMMS-21-23560-T

We thank the three Reviewers for their second round of thorough and insightful reviews of our manuscript. In particular, we are pleased to hear that Reviewer #3 supports the publication of this paper. Below, we address the comments from all three Reviewers on a point-by-point basis. We also revised the manuscript and highlighted the changes in blue fonts.

Response to Reviewer #1

1. *“I have carefully read the response letter, the modified manuscript, and the SI. While I see some improvements, my main concerns remain unresolved or vaguely addressed. I still find the manuscript interesting but identify several shortcomings in its current state, which I will delineate through my replies to the authors’ comments. In the present form, I cannot recommend this work for publication in Nature Communications.*

Response: We thank Reviewer #1 for the careful review of our revised manuscript. We appreciate the Reviewer for recognizing the interest of our work. We have addressed each of the Reviewer’s points below and in the manuscript.

2. *“Re: Response2 – Here, I meant S1 through S6 including the sub-numberings (S1-1, S1-2, S2-1, etc.). I have given another try at understanding the steps during this second revision and I still struggle to make a physical meaning out of the mathematical description in Eq. S6-2. The strength of harmonic SOT analysis lies in its general applicability to other similar systems of interest. This is best ensured if the developed method passes through the various sanity checks. I am curious if the authors have tried to set H_{DM} to zero and performed the simulations. Because in the absence of H_{DM} and H (or very small H), the system can be treated like two ferromagnets with opposite magnetization directions as far as H_{DL} is concerned. In this case, I strongly doubt that an in-plane tilting of the magnetization would occur because the opposite sublattices will act symmetrically to H_{DL} , which will tilt them out-of-plane in opposite directions but I do not see why an in-plane tilting would occur due to the antiferromagnetic coupling between the sublattices, unless there is a phenomenon that breaks this symmetry. When I set H_{DM} and H to (nearly) zero in Eq. S6-2 $\Delta\Phi$ term diverges to infinity, which is unphysical. Another sanity check is concerning the simulations. The authors have used a single set of parameters and performed the simulations for a fixed H_{DL} direction. What if you set $H_{DL} = -1$ Oe? Do you still get the same sign of $\Delta\Phi$ vs. Φ_H or the opposite sign? The same (opposite) sign would mean that this signal may show up in the third (second) harmonic. What about setting H_{DM} to zero? H_{DM} is specific to this material but it is not a general attribute to antiferromagnets. If the third harmonic response is somewhat linked to the presence of the DMI, some of the paper's claims (developing a general method to characterize SOTs) should be revisited. Nevertheless, unless significantly improved, even though fully clarified from the above ambiguities, I find the method presented here highly inaccessible to an average researcher who might be dealing with SOTs in antiferromagnets, let alone the more general audience.”*

Response: We thank Reviewer #1 for the excellent suggestions. Following these suggestions, we performed the corresponding simulations for damping-like torque, as shown in Figure R1 below.

Figures R1a and R1b are the simulations of $\Delta\theta_{A,B,n,m}$ and $\Delta\varphi_{A,B,n,m}$ when $H_{DL} = -1$ Oe. Compared with the simulations when $H_{DL} = 1$ Oe as shown in Figures S3c and S3d in the Supplementary Materials, $\Delta\theta_{A,B,n,m}$ reverses the sign while $\Delta\varphi_{A,B,n,m}$ remains the same, which is consistent with Eqs. S6-1 and S6-2. Furthermore, Figures R1c and R1d show the simulations of $\Delta\theta_{A,B,n,m}$ and $\Delta\varphi_{A,B,n,m}$ when $H_{DM} = 0$. The simulations still match well with Eqs. S6-1 and S6-2, indicating our model can be generalized to other antiferromagnets with zero H_{DM} .

Figures R1e and R1f show the field dependence of $\Delta\theta_{A,B,n,m}$ and $\Delta\varphi_{A,B,n,m}$ when $H_{DM} = 0$ and $\varphi_H = 45^\circ$. At $\mu_0 H > 0.2$ T, $\Delta\varphi_{A,B,n,m}$ follows $1/H$ dependence and $\Delta\theta_{A,B,n,m}$ remains constant, as predicted by Eqs. S6-1 and S6-2. However, at very small field, the expected divergence does not show up for $\Delta\varphi_{A,B,n,m}$, and both $\Delta\varphi_{A,B,n,m}$ and $\Delta\theta_{A,B,n,m}$ rapidly decrease to zero as field approaches zero. This is because the explicit solution of Eqs. S6-1 and S6-2 are valid only under the condition of $H_{FL}, H_{DL}, H_{ME} \ll H_K \ll H \ll H_{ex}$, which is not the case for small field. In reality, such unphysical divergence does not happen because of the multiple domain state at small field, and also because the minimum field we use is 0.3 T. Besides, the fitting curves always match quite well with the experimental data at magnetic field above 1 T when $\alpha\text{-Fe}_2\text{O}_3$ is in single domain state, while there is a small deviation for smaller fields. This also indicates that our model can correctly describe the current-induced effect in antiferromagnets, especially in single domain state.

The Reviewer points out that, “*in the absence of H_{DM} and H (or very small H), the system can be treated like two ferromagnets with opposite magnetization directions.*” When there is no H_{DM} and H , the large exchange field H_{ex} is the term that dominates antiferromagnetic dynamics. Since H_{ex} is much larger than all other terms, the tilting of two sublattice magnetizations is energetically unfavored. As a result, the system cannot be treated like two ferromagnets with opposite magnetization directions unless we set both H_{DM} and H_{ex} to 0. In that case, we obtain $\Delta\varphi = 0$ and $\Delta\theta = \frac{H_{DL}}{H_K} \sin \varphi_H$, which is the same as ferromagnets. **We totally agree with the Reviewer that in this case, in-plane tilting by damping-like torque would not happen.** The misunderstanding is due to lack of clarity in our manuscript. Actually, the in-plane tilting would not occur regardless of the existence of H_{DM} and H , as shown in Eqs. S6-1 and S6-2. We have added the following text to the Supplementary Materials (page 5) to clarify this point.

“As we can see from Eqs. S6-1 and S6-2, $\Delta\theta_A = -\Delta\theta_B$ and $\Delta\varphi_A = \Delta\varphi_B$. This is because the opposite sublattices act symmetrically to H_{DL} which tilts them out-of-plane in opposite directions. In-plane tilting would not occur due to the dominant antiferromagnetic exchange coupling between the sublattices. Mathematically, at equilibrium state, the two sublattice magnetization \mathbf{m}_A and \mathbf{m}_B stay in-plane antiparallel with each other. Under the damping-like torque where the effective field $\mathbf{H}_{DL,A(B)} \propto \mathbf{m}_{A(B)} \times \boldsymbol{\sigma}$, \mathbf{m}_A and \mathbf{m}_B tilt towards \mathbf{z} with $\Delta m_A^z = -\Delta m_B^z$. As a result, we have $\Delta\theta_A = -\Delta\theta_B$. Then, the out-of-plane component of sublattice magnetization experiences the other damping-like torque with $\Delta\mathbf{H}_{DL,A(B)} \propto \Delta\mathbf{m}_{A(B)} \times \boldsymbol{\sigma}$, which leads to $\Delta\varphi_A = \Delta\varphi_B$. Thus, there is no in-plane tilting.”

Figure R1. Simulations of angular dependence of $\Delta\theta(\varphi)_{A,B,n,m}$ under damping-like torque with **a, b**, $H_{DL} = -1$ Oe, and **c, d**, $H_{DM} = 0$. **e, f**, Simulations of field dependence of $\Delta\theta(\varphi)_{A,B,n,m}$ under damping-like torque with $H_{DM} = 0$, $\varphi_H = 45^\circ$. All other parameters are the same as those used in the Supplementary Materials.

3. “Re: Response3: This response, unfortunately, does not address my main concern that “the current method will fail to identify the torque direction”. They use a prototype SOT material

Pt from which we know the vector form of the SOTs and their approximate amplitudes. If the DL-SOT direction were to be characterized in a new material, the presented method would fail to do so since there is no vector information on the third harmonic signal. Not to mention the added complexity of having an accurate estimation of the ME contributions.”

Response: The Reviewer raised an excellent question. The scope of this work is to provide a method to detect antiferromagnetic dynamics. To the best of our knowledge, we are the first to investigate higher harmonics in such a prototype HM/AFM-insulator heterostructure. We chose Pt because it is the most commonly used SOT material.

In our follow-up studies, we will combine various antiferromagnets and other novel SOT materials. We agree that our modeling is based on heavy metals or other materials where the spin polarization is in-plane and perpendicular to the current direction. Here, we are happy to share our preliminary explorations with the Reviewer. Below we show some theoretical calculations and simulations if there exists a novel out-of-plane spin polarization σ_z . For the out-of-plane damping-like SOT (effective damping-like torque field $H_{DL,z}$), the changes of \mathbf{m}_A , \mathbf{m}_B , \mathbf{n} and \mathbf{m} are,

$$\Delta\theta_A = \Delta\theta_B = \Delta\theta_n = \Delta\theta_m = 0, \quad (\text{R1-1})$$

$$\Delta\varphi_A = \Delta\varphi_B = \Delta\varphi_n = \Delta\varphi_m = -\frac{2H_{\text{ex}}H_{DL,z}}{H(H+H_{DM})}. \quad (\text{R1-2})$$

Figure R2a shows the numerical simulation of coupled LLG equation, which matches Eqs. R1-1 and R1-2. Then the measured second harmonic signal is

$$V_{2\omega,DLz} = -V_{\text{TSMR}} \frac{2H_{\text{ex}}H_{DL,z}}{H(H+H_{DM})} \cos(2\varphi_H) \quad (\text{R2})$$

This is an intriguing result as out-of-plane damping-like torque appears in the second harmonic signal and has a unique angular dependence as compared with other components. Building on our current work, where the main goal is to show that the harmonic measurement is a powerful tool to detect current-induced effects in an antiferromagnet, this preliminary result further demonstrates that our method can be extended to more complex and novel SOT materials.

Figure R2. Simulations of **a**, $\Delta\theta_{A,B,n,m}$ and **b**, $\Delta\varphi_{A,B,n,m}$ under out-of-plane damping-like torque with $H_{DL,z} = 1$ Oe. All other parameters are the same as those in the Supplementary Materials.

4. “Re: Response4: OK, once the appearance of DL-SOT contribution to the third harmonic signal is clarified.”

Response: We are glad to hear that our response addressed the Reviewer’s comment.

5. “Re: Response5: I see and respect that the authors mainly rely on literature values of similar materials or exceptionally large values reported in Pd/Co multilayers to justify the large FL-SOT. First of all, the early findings in Pd/Co multilayers have been generally associated with spurious signals as recent work has failed to reproduce them (See: *Phys. Rev. Applied* 7, 014004 2017). The only original argument I can identify in the authors' response is that the antiferromagnets have distinct properties than the ferromagnets. This is surely the case, but without further intuitive explanation, this argument remains ambiguous and not satisfactory when explaining an important finding that the FL-SOT is ~ 35 times larger than the DL-SOT. With the current understanding of the spin-torque phenomenon, this is unphysical, or a fundamentally new understanding is at stake.”

Response: We thank Reviewer#1 for pointing out that the large FL-SOT found in Pd/Co multilayers may not be reliable. However, significantly larger FL-SOT compared with DL-SOT is not rare. For example, in *Phys. Rev. Mater.* 5, 045003 (2021), the authors find that in Ru₂Sn₃/CoFeB bilayers, FL-SOT is 30 times larger than DL-SOT. We know that FL(DL)-SOT is determined by the imaginary (real) part of spin mixing conductance g_i (g_r). It is widely accepted that in magnetic insulators such as Tm₃Fe₅O₁₂, Y₃Fe₅O₁₂ or EuS, much larger g_i than g_r can be achieved. For experiment, please see *Phys. Rev. Lett.* 111, 106601 (2013) and *Nano Lett.* 20, 6815–6823 (2020); for theory, see *Appl. Phys. Lett.* 117, 022404 (2020). This aligns with the difference between the interface of HM/Magnetic-Insulator and its metallic counterpart. Thus, the insulating property of α -Fe₂O₃ could lead to much larger FL-SOT based on similar physics. We agree that there is a lack of detailed theoretical studies of large FL-SOT in HM/AFM. Hopefully, our results will attract interest in our community, which is one of our goals for this work. We have added the following text to the main text to better address this point.

“One notes that this is an order of magnitude smaller than H_{FL} , which may be related to the insulating nature of α -Fe₂O₃. It is known that FL(DL)-SOT is determined by the imaginary (real) part of spin mixing conductance. In HM/ferromagnetic-insulator heterostructures such as Pt/Y₃Fe₅O₁₂ and Pt/EuS, the imaginary part of spin mixing conductance is an order of magnitude larger than its real part.³⁹⁻⁴¹ Further research in HM/AFM-insulator is needed to better understand the SOTs in AFM heterostructures”

6. “Re: Response6: I would be reluctant to accept that “this work shows that harmonic measurements can be smoothly generalized in HM/AFM systems and provides a powerful tool to the community to analyze all current-induced effects”. A significant improvement is required to reach this point. Performing measurements in different antiferromagnetic systems would be cumbersome at this stage. A suggestion is to perform simulations spanning a parameter space that would also be of interest to other common antiferromagnetic systems (NiO, Cr₂O₃, CoO, etc.) and showing/discussing them in the main text. This would be useful guidance for future

similar studies.”

Response: We thank the Reviewer’s suggestion here and totally agree that extending our work to other antiferromagnets is necessary. As the Reviewer mentioned, our work is mainly focused on a proof of demonstration for harmonic measurements in antiferromagnets. From the simulation side, our model can be directly used in other easy-plane antiferromagnets such as NiO and CoO when the field is within the easy plane. For the field along different crystal orientations, it is much more complicated. We are collaborating with theorists to develop a model and perform the measurements. We appreciate the Reviewer for understanding the difficulty of optimizing a different material heterostructures in a short period of time. Following the Reviewer’s suggestion, we performed some preliminary measurements on another antiferromagnetic insulator NiO. As shown in Figures R3a to R3c, we have observed some similar second and third harmonic results compared with those in α -Fe₂O₃, but with much smaller FL and DL-SOT as well as ME. This is reasonable since NiO needs much larger field to achieve single domain state. Thus, we might need to improve our signal-to-noise ratio as well as sample quality in the next round experiments in order to clearly separate those signals. We will definitely perform more comprehensive measurements and analysis in the future, which will likely take considerable amount of time and efforts.

Figure R3. In-plane angular dependence of harmonic Hall voltages **a**, $V_{1\omega}$, **b**, $V_{2\omega}$ and **c**, $V_{3\omega}$ for a Pt(5 nm)/NiO(111)(20 nm) bilayer at different magnetic fields from 0.5 to 14 T at 300 K.

7. -There is an inconsistency in the manuscript regarding the use of data at low and high fields to characterize the DL-SOT and FL-SOT. The authors rely on the high field values for the characterization of the FL-SOT due to possible signal contamination at low fields from the multidomain state. However, the $\sin(4\phi)$ signal in the third harmonic channel, presumably responsible for the DL-SOT, only appears at and below 1T with a magnitude that could be unequivocally associated with a signal having a physical origin. Above this field value, the

signal is a carbon copy of the first harmonic signal, which is understood as the resistivity term (not of SOT origin). It is not possible to use harmonic analysis for materials in a multidomain state because of the unavoidable current-induced domain wall motion generated harmonic signals that would overlap the actual signal. The fact that the DL+ME terms are absent in the single domain state casts further doubt about the real origin of the third harmonic signal in this sample.

-In response to Reviewer1 the authors perform current-dependence of the 2nd and 3rd harmonic measurements and show them in the SI. It seems like at 1 mA no signal is present at all and at higher currents, the expected dependencies are not always strictly observed. The authors should comment on these inconsistencies.

Response: We understand Reviewer #1's concern on the multidomain state. However, we would like to point out that although a simply glance at Figure 3a implies the $\sin 4\phi_H$ signal in the third harmonic channel only appears at ≤ 1 T, the extracted data in Figure 4a clearly shows the $\sin 4\phi_H$ term exists up to 7 T. We agree that the unavoidable current-induced domain wall motion affects the field-like torque magnitude at small field, but it has much less impact on DL and ME. As can be seen in Fig. 4 of the main text, the fitting curve matches well with the experimental data at ≥ 1 T when the α -Fe₂O₃ film is in single domain state, while at small field below 1 T only slight deviation is observed. This is because according to Fig. 1e, even at 0.3 T, there are still more than 75% of the Néel vectors that have been aligned by the external field. Thus, although the measured $\sin 4\phi_H$ signal is less than that predicted by the model based on perfect single domain state, the impact of multidomain state does not change the qualitative behavior of SOTs.

For the current dependent measurement at 1 mA, the signal-to-noise ratio is so low that we could not extract any meaningful information. However, this should not mean there is zero signal at 1 mA. From the fitting curves in Fig. S7, we can see that the signal at 1 mA should be very small. Thus, our current-dependent measurements are still consistent with the theoretical prediction. For example, the spin Seebeck signal in Fig. S7a is expected to have a quadratic current dependence as the fitting curve indicates. Meanwhile, the experimental data in Fig. S7a appear to have a linear dependence, most likely because the data point at 1 mA is set to zero due to experimental sensitivity, which should have a small but non-zero value. If we only consider the data at 2, 3 and 4 mA, we find that the value of $V_{2\omega, SSE}/H$ is 0.7, 1.5, and 2.4×10^{-8} V/T, respectively. This is closer to a quadratic dependence instead of a linear current dependence. Figure R4 shows these data and the corresponding quadratic and linear fits, where the quadratic fit passes through origin while the linear fit results in negative $V_{2\omega, SSE}/H$ values for 0 and 1 mA, which is unphysical. We hope to convince the Reviewer that the zero signal at 1 mA is because of the limit of our experimental sensitivity, and the SSE component most likely follows a quadratic current dependence as expected from our modeling.

Figure R4. Spin Seebeck effect coefficient in a Pt(5 nm)/ α -Fe₂O₃ bilayer as a function of applied current (2, 3, and 4 mA) measured at 300 K, which is fitted by a quadratic (blue) and linear (green) function.

Response to Reviewer #2

1. “I thank the authors for considering my comments/criticisms in their modified draft.
1) However, the answer of the authors regarding my main critique 1 is unfortunately not satisfying and my doubt about the symmetry of the second harmonic component of the transverse voltage signal remains.

As the authors correctly note: for strong applied magnetic fields above the spin-flop transition where the Neel vector aligns perpendicular to the magnetic field for all field angles, the second harmonic signal generated by the field-like torque follows a $\cos(\phi_H)$ dependence in their model and therefore is “odd under reversal”.

However, in my opinion, the symmetry of the second harmonics should also follow an “odd under field reversal” angular dependence at small applied field strengths, even if axial and/or easy-plane anisotropies prevent the full Neel vector from being oriented perpendicular to the applied field. Thus, I still cannot understand how a $\cos(2\phi_H)$ can appear as the dominant component of the measured second-harmonic signal even in a fictitious multidomain state at low fields, since all contributions from all the individual domains should have no or an “odd” field dependence due to the field induced tilt of the Néel vector. Since the origin of this largest of all higher harmonic signal is not traced, doubts arise also about the interpretation of the much smaller third harmonic signal.

The authors should therefore replace the vague statement that the angular dependence is “complicated and uncertain” by a comprehensible interpretation of the even under field reversal contribution of the second harmonic signal.”

Response: We are grateful to Reviewer #2 for acknowledging our previous revision. From the symmetry point of view, we totally agree with the Reviewer that *all the individual domains should have **no or an “odd”** field dependence due to the field induced tilt of the Néel vector.* If there is a field-independent term (in fact, we only need field orientation independence) due to the field-induced tilt of the Néel vector, the generated second harmonic signal has the form of $\cos(2\phi_H)$. This is because $V_{2\omega} \propto \frac{dV_{1\omega}}{dI}$, where $V_{1\omega} \propto \sin 2\phi_H$ and $V_{2\omega} \propto \cos 2\phi_H \Delta\phi_n$. Thus, if $\Delta\phi_n$ has no field dependence, $V_{2\omega} \propto \cos 2\phi_H$.

It should be pointed out that this even term in the second harmonic signal only dominates at low field in multidomain state, and has no counterparts at the third harmonic channel. As shown in Fig. 4, the data only slightly deviates from the fitting curve when the field is below 1 T, whereas it deviates a lot for field-like torque in Fig. 2e. This also implies that the even term is likely originated from the field-like torque in multidomain state which does not appear in the third harmonic voltage. Furthermore, the magnitude of the third harmonic voltage is of the same order of the second harmonic voltage, only slightly smaller. Therefore, the even term should not affect the analysis of third harmonic signal, which is also an advantage of harmonic characterization.

To better understand the origin of this term, we have conducted measurements in many Pt/Fe₂O₃ samples to verify the reproducibility. We find that all the other terms can be reproduced except for this even term which does not always show up at small fields. This makes us believe that this even term is highly sensitive to the initial domain state since the size and configuration of antiferromagnetic domains in Fe₂O₃ after growth is rather random. We would appreciate the

Reviewer's thoughts on this aspect.

In addition, in our Supplementary Materials, we added another possible explanation (on page 12, also copied below) related to the spin Seebeck effect that depends on the Néel order as recently reported by Ross, *et al.* (*arXiv:2105.13653*) in Pt/ α -Fe₂O₃.

“If there is a field orientation independent contribution to $\frac{d\varphi_n}{dt}|_{I=0}$ in Eq. S10-1, which is allowed from the symmetry point of view, there could be a second harmonic term that is even to the magnetic field.”

2 “2) Current-amplitude dependence of the signal of the 2nd second harmonic:

The authors now show in Fig. S7(a) the current dependence of the signal contribution attributed to the spin Seebeck effect, i.e., $\sim \cos(\phi_H)$. Surprisingly, the current dependence seems to follow a linear rather than the expected quadratic dependence.

Certainly very illuminating with respect to the interpretation of the harmonic signals is the current dependence of the 2nd harmonic contributions attributed to the field-like torque, which include both the "odd" term $\sim \cos(2\phi_H) \cos(\phi_H)$ and the questionable "even" term $\sim \sin^2(\phi_H)$.

I would be very interested to see the current dependence of these two contributions both at low field below the spin flop transition and at high field above the transition..”

Response: The spin Seebeck signal in Fig. S7a is expected to have a quadratic current dependence as the fitting curve indicates. It might look like a linear dependence because the signal at 1 mA is below our measurement noise level and was set to zero; but it should have a small but non-zero value. If we only consider the data at 2, 3 and 4 mA, we find that the value of $V_{2\omega, SSE}/H$ is 0.7, 1.5, and 2.4×10^{-8} V/T, respectively, which is closer to quadratic dependence instead of linear current dependence. As shown in Fig. R4 above with both the quadratic and linear fits, the quadratic fit passes through origin while the linear fit results in negative $V_{2\omega, SSE}/H$ values for 0 and 1 mA, which is unphysical.

The fitting of $V_{2\omega, FL}$ and $V_{2\omega, FL}'$ based on high field data has been shown in Figs. S7b and S7c. However, for low field region, we cannot use our model to fit because our model is based on single domain state. Figures R5a and R5b below show the field dependence of $V_{2\omega, FL}$ and $V_{2\omega, FL}'$ at 2, 3 and 4 mA. Figures R5c and R5d show current dependence of $V_{2\omega, FL}$, and $V_{2\omega, FL}'$ at 0.3 T, 0.5T and 1 T. It also follows quartic current dependence like the high field branch. We would be grateful to the Reviewer's insights regarding this point.

Figure R5. Second harmonic results. Field dependence of **a**, $V_{2\omega,FL}$ and **b**, $V_{2\omega,FL}'$ at currents of 2, 3 and 4 mA, where the insets of **a** and **b** show the corresponding $1/H$ plots and linear fitting. Current dependence of **c**, $V_{2\omega,FL}$ and **d**, $V_{2\omega,FL}'$ contributions at low fields with a quadratic current dependence.

3 “3) I thank the authors for explaining to me in detail how H_{FL} , H_{DL} , and H_{ME} were determined.”

Response: We are glad that our response addressed the Reviewer’s comment.

4 “4) As a minor point: to achieve current-induced switching of the AF order and detection of the 2nd harmonic of the 180 degree inversion, the requirement of broken space inversion symmetry is also realized at an AF/HM interface and bulk space inversion symmetry of the AF is not necessarily required I think.”

Response: We agree with the Reviewer that broken space inversion symmetry at AF/HM interface makes 180° Néel order reversal possible to be detected in harmonic measurement even that AF preserves bulk space inversion symmetry. Although so far there is no reported work yet, it would be interesting to explore its possibility.

5 “5) and 6): I agree with the authors, thank you.”

Response: We thank the Reviewer for evaluating our response.

6 “In summary, I think that the problem with the symmetry of the 2nd harmonic signals, which is important, has not yet been solved in the current draft of the paper. Perhaps the analysis with respect to the current dependences will help identifying the origins of the different 2nd harmonic terms.”

Response: We agree with the Reviewer that fully understanding the 2nd harmonic signals is important, while the main focus of our paper is the 3rd harmonic signals. As we mentioned previously, one of the merits of harmonic measurements is that the contributions from different orders are separated. That mystery/questionable term in the 2nd harmonic signals should be further investigated, but should not directly affect the analysis we performed in the 3rd harmonic terms.

Response to Reviewer #3

1. “I was a reviewer of this paper at the previous round (before it was transferred to Nature Communications. I have read the authors' response to all the comments (mine as well as other Reviewers) and find them satisfactory. Moreover, after reading the comments of my colleagues I agree that the paper is more suitable for Nature Communications and would recommend to publish it in the present form.”

Response: We thank Reviewer #3 for recognizing the value of our work. We thank all the comments and suggestions from the Reviewer which greatly improve the quality of our work.

Reviewers' Comments:

Reviewer #1:

Remarks to the Author:

The authors have addressed my previous concerns and criticism to a great extent. I am now supportive of the publication of this manuscript in Nature Communications after some minor revisions listed below:

- Unlike the authors, I do not believe that the much larger imaginary part of the spin mixing conductance (with respect to the real part) in magnetic insulator/metal interfaces is a widely accepted fact. There are some such examples but, this cannot be generalized. The authors should revise the corresponding statement.

- I still believe that the text is poorly accessible to the Nature Communications readership due to the inherent/unavoidable complexity of antiferromagnetic magnetization dynamics and related equations. The authors could make an additional effort to simplify their language and their expressions (especially the use of subscripts, angle definitions, etc.) and maybe use more illustrative figures.

Reviewer #2:

Remarks to the Author:

After reading the answers to my questions and also to the questions of reviewer 1, I am unfortunately even more confused than I was already after reading through the original manuscript. It is evident from the present work that there are undoubtedly nonlinear magnetoresistance responses measured at various harmonics, but the physical meaning of the various contributions is not adequately explained, at least it does not become clear to me. For example, as mentioned in my previous reports, it is unsatisfactory to attribute the largest of all (2nd) harmonic contribution with an angular dependence that contradicts the model to a fictitious low field multidomain state without a physical explanation of how such an assumed multidomain state might produce this unexpected angular dependence.

Lastly, the system under study also lacks overall generality. It is at least not more general than the "AFM with broken time reversal and space inversion symmetry". All measurements on the bi-layer system described here with Dzyaloshinskii-Moriya interaction (space inversion symmetry...?) were performed under the influence of a (strong) magnetic field (time irreversal symmetry....?).

In summary, I can not encourage the publication of this work because both of the persisting ambiguities in the analysis of all higher harmonic signals and also because of the lack of generality.

Second Response to reviews on manuscript NCOMMS-21-23560-T

We thank the Reviewers for their second round of insightful reviews of our manuscript. In particular, we are pleased to hear that Reviewer #1 is now supportive of the publication of this manuscript in Nature Communications. We also appreciate Reviewer #3's support of this paper. Below, we address the comments from the Reviewers on a point-by-point basis. We also revised the manuscript and highlighted the changes in blue.

Response to Reviewer #1

1. *"The authors have addressed my previous concerns and criticism to a great extent. I am now supportive of the publication of this manuscript in Nature Communications after some minor revisions listed below:"*

Response: We appreciate Reviewer #1 for recommending the publication of this paper. We are grateful for the comments and suggestions from the Reviewer #1, which have significantly improved the clarity of this paper.

2. *"Unlike the authors, I do not believe that the much larger imaginary part of the spin mixing conductance (with respect to the real part) in magnetic insulator/metal interfaces is a widely accepted fact. There are some such examples but, this cannot be generalized. The authors should revise the corresponding statement."*

Response: Following Reviewer #1's suggestion, we clarify the large imaginary part of the spin mixing conductance in magnetic-insulator/metal interfaces by adding an additional section in Supplementary Materials as shown below.

"6) Large imaginary part of spin mixing conductance in HM/magnetic-insulator heterostructures

The spin-orbit torque can be expressed as,⁴

$$\tau_{\text{SOT}} = \tau_{\text{DL-SOT}} + \tau_{\text{FL-SOT}} = \frac{\hbar}{2e} [G_{\text{r}} \mathbf{m} \times (\mathbf{m} \times \boldsymbol{\sigma})] + \frac{\hbar}{2e} [G_{\text{i}} \mathbf{m} \times \boldsymbol{\sigma}] \quad (\text{S19})$$

Here, $G_{\text{r(i)}}$ is the real (imaginary) part of spin mixing conductance which determines the magnitude of DL(FL)-SOT. In HM/metallic-ferromagnet systems, $G_{\text{r}} \gg G_{\text{i}}$, which leads to a much larger H_{DL} compared with H_{FL} . This is confirmed by both experiments and first-principle calculations.^{31, 32} However, in HM/magnetic-insulator systems which receive much attention recently, this is not always the case. In Pt/Y₃Fe₅O₁₂ (YIG) and Pt/EuS bilayers,³³⁻³⁵ much larger G_{i} than G_{r} has been reported. Large H_{FL} in our Pt/ α -Fe₂O₃ might also be related to the insulating property of α -Fe₂O₃. To date, however, there are only few works that calculated the spin mixing conductance in HM/magnetic-insulator bilayers,³⁶ but failed to match the recent experimental results shown above. Further research in HM/magnetic-insulator systems is required address this question."

3. *I still believe that the text is poorly accessible to the Nature Communications readership due to the inherent/unavoidable complexity of antiferromagnetic magnetization dynamics and*

related equations. The authors could make an additional effort to simplify their language and their expressions (especially the use of subscripts, angle definitions, etc.) and maybe use more illustrative figures.

Response: We agree with Reviewer #1 that the manuscript should be as clear as possible to the readership, especially for the new regime of higher harmonics in antiferromagnets. We acknowledge that besides the experimental results, we have devoted significant efforts in modeling the current-induced effects in antiferromagnets, and consequently, making the expressions and discussions lengthy. To improve the readability, we put mathematical derivations and detailed discussion of the modeling in the Supplementary Materials and focus on the experimental results in the main text. We hope that for readers who are primarily interested in the experimental results of current-induced effects in antiferromagnets, the main text should be sufficient without going into details of the modeling. For readers who are interested in the modeling, our Supplementary Materials provides the details.

In addition, although the dynamics of antiferromagnets is complicated, the basic mathematical derivation is similar to those of ferromagnets. We modified the Supplementary Materials to help readers better understand the differences and similarities between antiferromagnets and ferromagnets. For example, in the last round of review, Reviewer#1 pointed out the lack of clarify about *in-plane tilting of the magnetization*. Following this comment, we revised the manuscript to clarify the dynamical processes in antiferromagnets, which hopefully will help the readers build the physical picture of antiferromagnetic dynamics that is dominated by the exchange field. This time, we make modifications to further improve the readability. For example, we add phrases such as “which are similar to ferromagnets” in several places to build the connection between antiferromagnets and ferromagnets. This should facilitate the understanding of our results and model since readers are generally more familiar with ferromagnetic dynamics.

As suggested by the Reviewer, we also tried to make the mathematical terms easier to read and understand. In particular, we add descriptions of the polar and azimuthal angles of the vectors in the captions of Figs. 1b-1c in the main text and Fig. S2 in the Supplementary Materials to help the readers understand the meanings of the mathematical symbols. We also change the mathematical terms with lengthy, multi-part subscripts to a new format using both subscripts and superscripts, such as from $V_{2\omega,FL}$ to $V_{2\omega}^{FL}$, through the manuscript and Supplementary Materials (including figures), which hopefully will be slightly easier to read. We hope our modifications can improve the readability of our paper.

Response to Reviewer #2

1. *“After reading the answers to my questions and also to the questions of reviewer 1, I am unfortunately even more confused than I was already after reading through the original manuscript”*

Response: We thank Reviewer #2 for the careful review of our revised manuscript. Having cleared the concerns from Reviewer #1 in the last round of review, we hope that we will be able to address the concerns from Reviewer#2. Most importantly, we appreciate the valuable communications with the Reviewers.

2. *“It is evident from the present work that there are undoubtedly nonlinear magnetoresistance responses measured at various harmonics, but the physical meaning of the various contributions is not adequately explained, at least it does not become clear to me. For example, as mentioned in my previous reports, it is unsatisfactory to attribute the largest of all (2nd) harmonic contribution with an angular dependence that contradicts the model to a fictitious low field multidomain state without a physical explanation of how such an assumed multidomain state might produce this unexpected angular dependence.”*

Response: We thank Reviewer #2 for raising this important question. From our understanding, a physical explanation of how the multidomain state might produce such an angular dependence consists of two parts:

- 1) Qualitative explanation on why it could produce an even-field angular dependence;
- 2) Quantitative explanation on why it could produce exactly $\sin^2\varphi_H$ angular dependence.

For the first part, we have added a section in the Supplemental Materials to elaborate this point based on our discussions with Reviewer#2 using symmetry analysis. This is similar to how a previous paper demonstrates the appearance of anisotropic magnetoresistance (AMR) in the 2nd harmonic measurement on antiferromagnetic CuMnAs films as reported in:

- J. Godinho, H. Reichlová, D. Kriegner, V. Novák, K. Olejník, Z. Kašpar, Z. Šobáň, P. Wadley, R. P. Campion, R. M. Otxoa, P. E. Roy, J. Železný, T. Jungwirth & J. Wunderlich, *Nat. Commun.* 9, 4686 (2018).

As the authors claimed in that *Nat. Commun.* work, *“Symmetry arguments are the basis for analyzing whether a given effect can in principle exist in a certain class of materials”*.

For the second part, one notes that the complexity of multiple domain state in antiferromagnets makes it difficult to build a comprehensive model. We have proposed several possibilities including field-like torque and Néel order related spin Seebeck effect. We totally agree with the Reviewer that this is not the most satisfactory way to address this term at low field regime, and that our model cannot explain the multidomain state quantitatively. Because our work is mainly a proof of principle that higher order (2nd and 3rd) harmonic measurements can be utilized to analyze current-induced effects in antiferromagnets, our model is based on the comparatively simpler case of single domain state. Actually, our model matches our experimental results quite well at field larger than 1 T when the α -Fe₂O₃ film is in the single domain state.

We would also like to point out that *the largest of all (2nd) harmonic contribution* is the largest only at *low field multidomain state*. Thus, we believe that the lack of quantitative modeling at low fields (multidomain state) should not undermine the success of modeling at high fields (single domain state) in our work.

Antiferromagnetic spintronics is a fast-developing, promising area in spintronics research, and there are many remaining questions that need to be answered. One of those is lacking an appropriate model to describe the multidomain behavior. Thus, we respectfully disagree with Reviewer#2 that this term *“contradicts the model to a fictitious low field multidomain state”* since there is no such a model to quantitatively describe the low field multidomain state; to build such a model is beyond the scope of this work. Given that there are numerous intriguing phenomena in this emerging field, we truly believe that our work will attract significant interest in the research community and inspire both theorists and experimentalists to explore the rich behaviors, including the multiple domain state, of antiferromagnetic heterostructures.

3. *“Lastly, the system under study also lacks overall generality. It is at least not more general than the “AFM with broken time reversal and space inversion symmetry”. All measurements on the bi-layer system described here with Dzyaloshinskii-Moriya interaction (space inversion symmetry...?) were performed under the influence of a (strong) magnetic field (time irreversible symmetry....?). In summary, I can not encourage the publication of this work because both of the persisting ambiguities in the analysis of all higher harmonic signals and also because of the lack of generality.”*

Response: First, we would like to clarify that it is true that α -Fe₂O₃ has intrinsic Dzyaloshinskii-Moriya interaction that indicates the broken space inversion symmetry. However, the experimental result shows that DMI only has a small modification to our model in the third harmonic terms, which matches with our model quite well. This indicates that all experimental results observed in our α -Fe₂O₃ do not require broken time reversal and space inversion symmetry, unlike the antiferromagnetic CuMnAs films in *Nat. Commun.* 9, 4686 (2018). This means our model can be used to analyze current-induced effects in all AFMs with or without broken time reversal and space inversion symmetry. As for the influence of a magnetic field, it is true external field breaks the time reversal symmetry. However, this is not related to the intrinsic property of AFMs, but the experimental method. When we discuss about “*generality*”, it refers to that this method (higher harmonic measurement) should work for all AFMs or certain kinds of AFMs. Harmonic measurements, including both FMs and AFMs, generally need a magnetic field to align their magnetic order. The harmonic technique has already been proven to be a powerful way to detect FM dynamics, while our work demonstrates its effectiveness in AFMs with or without time reversal and space inversion symmetry. Since our work is the first higher harmonic measurement (2nd and 3rd) HM/AFM-insulator bilayers, the exact scope of generality for this technique requires exploration of many other AFMs.

Reviewers' Comments:

Reviewer #1:

None

Reviewer #2:

Remarks to the Author:

The physical interpretation presented in this paper relies solely on the analysis of magnetotransport measurements, without presenting additional independent experimental evidences (such as the identification of antiferromagnetic domain structures, e.g., by magneto-optical measurements) that could support the interpretation of the measured electrical signals. Therefore, I would at least expect that all detected magnetotransport signals measured at different harmonics are consistent with the explanatory model proposed in this work.

Unfortunately, this is not the case for the largest measured higher harmonic signal, the 2nd harmonic signal measured at low fields.

In Fig. 2 of the main text and in the 4th paragraph of the supplementary materials, the authors attribute the measured 2. Harmonic signal to (1) a spin-Seebeck effect term $\sim \cos(\phi_H)$ generated by Joule heating, (2) a field-like torque-driven term $\sim \cos(2\phi_H)\cos(\phi_H)$, and (3) a hypothetical field-like torque term $\sim \sin(2\phi_H)$ acting on a hypothetical multidomain state.

While the first two terms are odd under field reversal and are plausible in their physical interpretation, the largest term (3) is even under field reversal and lacks a physical interpretation: How can field like torque or Joule heating acting on a multiple domain state produce an even $\sim \sin(2\phi_H)$ signal if each of the individual (single) domains contributes only terms that are odd under field reversal?

Also, the cited paper by Ross et al. describing the spin Seebeck effect in antiferromagnetic hematite does not include a symmetry analysis with respect to the field angle dependence of the signals measured there.

However, as I understand it, Ross et al. assigns the Seebeck signal measured above the spin-flop field to a contribution, which is proportional to the field-induced magnetization.

The signal measured at weak applied fields below the spin-flop field are assigned to a spin-Seebeck effect from Néel order, which results from the field induced unequal population of otherwise degenerate magnonic modes. In contrast to the 2nd harmonic signal described in the present paper by Cheng, et al., the Seebeck signals above and below the spin flop field, change their sign when the magnetic field reverses.

In summary, the authors' interpretation of the origin of the second harmonic signal remains unconvincing and questions the correctness of the other contributions of higher harmonics.

For that reason which I already mentioned in my previous reports, I cannot support the publication of this work.

Third Response to reviews on manuscript NCOMMS-21-23560-T

Response to Reviewer #2

1. *“The physical interpretation presented in this paper relies solely on the analysis of magnetotransport measurements, without presenting additional independent experimental evidences (such as the identification of antiferromagnetic domain structures, e.g., by magneto-optical measurements) that could support the interpretation of the measured electrical signals. Therefore, I would at least expect that all detected magnetotransport signals measured at different harmonics are consistent with the explanatory model proposed in this work. Unfortunately, this is not the case for the largest measured higher harmonic signal, the 2nd harmonic signal measured at low fields.*

Response: We thank Reviewer #2 for the third round of insightful reviews of our manuscript. We fully understand the concerns about the second harmonic signal measured at low field. The multiple domains at low field cannot be explained by the simplified model assuming the AFM is in the single domain state. We agree that introducing an even $\sin^2(\varphi_H)$ term is not appropriate, which could confuse the readers and weaken the robustness of our work. To address Reviewer #2’s concerns, we have substantially revised our analysis and the details are given below in our response to Comment 2.

Magneto-optical measurements, such as quadratic MOKE, have been used to image AFM domains. For example, an excellent work by H. Meer *et al.* (Nano Lett. 2021, 21, 114–119) used MOKE imaging to identify AFM domain switching in NiO. However, to date, there are few AFMs that have large quadratic MOKE signals as in NiO. We have tried MOKE imaging on our Fe₂O₃ thin films using a MOKE setup similar to that used in the work by H. Meer *et al.*, but did not see clear AFM domains. This could be due to the submicron AFM domain sizes in our Fe₂O₃ films, which is beyond the spatial resolution of MOKE imaging [see our recent work of XMLD-PEEM imaging on our Fe₂O₃ films in E. Cogulu, *et al.* Phys. Rev. B 103, L100405 (2021)]. We point out that in harmonic measurements, there is no AFM domain switching. Unlike the switching of AFM domains, harmonic measurements use an AC current to make the AFM spins slightly deviate from their equilibrium orientations. Such vibration of magnetization around the equilibrium orientations can be detected using electric transport measurement, whereas it is impossible to probe through image techniques such as MOKE or XMLD.

2. *In Fig. 2 of the main text and in the 4th paragraph of the supplementary materials, the authors attribute the measured 2. Harmonic signal to (1) a spin-Seebeck effect term $\sim \cos(\varphi_H)$ generated by Joule heating, (2) a field-like torque-driven term $\sim \cos(2\varphi_H)\cos(\varphi_H)$, and (3) a hypothetical field-like torque term $\sim \sin(2\varphi_H)$ acting on a hypothetical multidomain state.*

While the first two terms are odd under field reversal and are plausible in their physical interpretation, the largest term (3) is even under field reversal and lacks a physical interpretation: How can field like torque or Joule heating acting on a multiple domain state produce an even $\sim \sin(2\varphi_H)$ signal if each of the individual (single) domains contributes only terms that are odd under field reversal?

*Also, the cited paper by Ross *et al.* describing the spin Seebeck effect in antiferromagnetic*

hematite does not include a symmetry analysis with respect to the field angle dependence of the signals measured there.

However, as I understand it, Ross et al. assigns the Seebeck signal measured above the spin-flop field to a contribution, which is proportional to the field-induced magnetization.

The signal measured at weak applied fields below the spin-flop field are assigned to a spin-Seebeck effect from Néel order, which results from the field induced unequal population of otherwise degenerate magnonic modes. In contrast to the 2nd harmonic signal described in the present paper by Cheng, et al., the Seebeck signals above and below the spin flop field, change their sign when the magnetic field reverses.

Response: We greatly appreciate the discussions with Reviewer #2 on the physical origin of the even $\sin^2(\varphi_H)$ term. The multiple domain scenario at low fields is beyond the scope of our model that assumes a single domain state. We agree that it is inappropriate to introduce such an additional term that cannot be well explained. Thus, we have **removed the $\sin^2(\varphi_H)$ term** in our fitting of the second harmonic data. We replot the figures [Figure 2 in the main text and Figures S6-S8 in the Supplementary Information] and delete the relevant discussions of this term in the main text and the Supplementary Information. The new analysis shows that removing the $\sin^2(\varphi_H)$ term only leads to very minimal change ($< 3\%$) of the magnitude of spin Seebeck effect and field-like torque. We notice that overfitting the data in the low field regime will be likely the “*Drawing an Elephant with Four Parameters*”, which could confuse the reviewers and the readers.

3. *In summary, the authors' interpretation of the origin of the second harmonic signal remains unconvincing and questions the correctness of the other contributions of higher harmonics. For that reason, which I already mentioned in my previous reports, I cannot support the publication of this work.”*

Response: We thank Reviewer #2 for the careful review of our revised manuscript. We have tried our best to address Reviewer#2's concerns. We especially thank Reviewer#2 for pointing out the inappropriateness of introducing an additional term in the second harmonic measurement without a convincing explanation. We believe that due to the different origins of the second and the third harmonic signals, and the excellent fitting of our data in the third harmonic measurement, our analysis is convincing now. We hope this could clear the concern from the Reviewer and get the support for publication.

Reviewers' Comments:

Reviewer #2:

Remarks to the Author:

The authors now disregard their previously determined " $\sin(2\phi_H)$ " contribution in the second harmonic response by fitting the measured data to a fitting formula containing only terms that are plausible in their model. In my opinion, this is not fully satisfactory, especially since a nonlinear response corresponding to a higher harmonic order measurement is directly related to the measured response at the lower harmonic order. A better way would be to clarify in a comprehensible way the origin of a term that is inconsistent with a simplifying model rather than ignoring it.

Although the authors mention in their response letter that they were unable to image AFM domains with the quadratic MOKE, they do, on the other hand identify in their PRB 103, L100405 (2021) small AFM domains in comparable Pt-coated α -Fe₂O₃ films of the same nominal thickness using XMLD scans. In this previous work, they even show (nonvolatile) current-induced switching of AFM domains with probably the same current-induced effective fields as they use here at lower current densities to produce the small Néel vector deviations responsible for the higher harmonic signals. Therefore, it would have been indeed more convincing for me if the higher harmonics measurements had been applied as well to the identification and analysis of permanently switched states. Because in this case, this method would have been applied to an undisturbed AFM system that is not affected by a very strong magnetic field of up to 14 T, which generates an additional net magnetization in the AFM state and possibly magnetizes the strongly magn-field susceptible platinum layer.

This leads me to my last point, which I actually mentioned already in my very first report. The sentence in the abstract "The harmonic measurement technique has never been verified in an antiferromagnetic heterostructure" is incorrect, since harmonic measurement techniques have in fact already been used in an antiferromagnetic heterostructure, namely Pt/CuMnAs, to identify 180-degree current-induced Néel vector switching (Nature Communications 9, 4686 (2018)). Here, the second harmonic signal is the result of both a small current-induced spin-orbit field capable of deflecting the Néel vector away from its equilibrium position of switched AF states.

In my opinion, it is basically possible to characterize any AFM heterostructure by a second harmonic response in which a staggered current induced effective field is able to deflect the Néel vector from its equilibrium positions and in which these equilibrium positions are not along extrema with respect to the relevant linear magnetotransport response.

Therefore, a second harmonic signal could in principle also be generated by a current-induced effective field corresponding to a damping-like torque in contrary to what is said in this paper.

In summary, I still feel that the present work characterizing the 3rd harmonic response to an antiferromagnetic heterostructure, without having fully clarified the 2nd harmonic response, is neither conclusive nor novel enough to be published in Nature Communications.

However, since I am repeating previous criticisms, perhaps the opinion of an additional reviewer should make the final decision on whether or not this work should be published in Nature Communications.

Reviewer #4:

Remarks to the Author:

Referee report on the manuscript "Third Harmonic Characterization of Antiferromagnetic Heterostructures" by Yang Cheng et al.

The manuscript deals with the harmonic characterization of antiferromagnetic materials to identify an electrical approach to characterize the damping-like torque and thermally-induced magnetoelastic effect contributions considering the third harmonic voltage. The material platform is the Pt/ α -Fe₂O₃. The manuscript is well written and the topic is timely. I have look into the calculations of the second harmonics voltage and the current description is correct. It captures the essential physics the authors wish to describe. I also do not think it is necessary to image the antiferromagnetic state and I agree with the response of the authors to the referee #1 point 1. On

the other hand, I noticed that the authors do not consider thermal effect on the magnetic parameters, for example the amplitude of field and damping like torque have a temperature dependence linked with the magnetic parameters of the antiferromagnets. This aspect has to be commented and checked.

In conclusion, I support the acceptance of the publication, however I have also additional minor comments listed below which should be discussed/addressed before publication.

I have the following comments on the manuscript. In the introduction several important work on antiferromagnets are not considered. First, the switching has been demonstrated in heavy-metal (HM)/AFM-metallic bilayers, and in those systems it has been also shown the scalability of the switching process in confined geometry and its potential compatibility with CMOS technology [see for example Shei et al Nature Electronics 3, 92–98 (2020)].

From what I found in literature, the debate on the mechanism of the Néel order switching, which could be induced by spin-orbit torque (SOT) or the magnetoelastic effect is important when considering AFM isolating systems while for metallic systems the debate is related to the contribution of the HM in the resistance change [Meinert et al, Phys. Rev. Applied 9, 064040, 2018,] and the effect of morphological features [Bodnar, et al, Phys. Rev. B 99, 140409(R), 2019]. The authors should expand more on this motivation.

May the authors add references to support the following sentence: "Whether harmonic measurement can be used in characterizing the current induced effect in other AFMs is still an open question."

Page 4, to fix Fig. d in Fig. 1d.

The authors cite the Supplementary Materials in the following sentence "which is the same as the transverse spin Hall magnetoresistance (TSMR) in DC measurements (see Supplementary Materials for more details)", I'm not sure to which part they are referring.

When discussing about "the effective anisotropic field of the magnetoelastic effect is induced by thermoelastic stress $\Delta\sigma$ " the authors should cite O Gomonay and D Bossini J. Phys. D: Appl. Phys. 54, 374004, 2021 where a complete theory has been developed in this context.

We thank Reviewer#2 for the fourth round of insightful reviews of our manuscript and for the suggestion of an additional reviewer to evaluate this manuscript. We also appreciate Reviewer#4 for supporting the publication of our work. Below we address the comments from all three Reviewers on a point-by-point basis. We also revised the manuscript and highlighted the changes in blue fonts.

Response to Reviewer #2

1. *The authors now disregard their previously determined " $\sin(2\phi_H)$ " contribution in the second harmonic response by fitting the measured data to a fitting formula containing only terms that are plausible in their model. In my opinion, this is not fully satisfactory, especially since a nonlinear response corresponding to a higher harmonic order measurement is directly related to the measured response at the lower harmonic order. A better way would be to clarify in a comprehensible way the origin of a term that is inconsistent with a simplifying model rather than ignoring it.*

Response: We understood Reviewer#2's concern and we have tried our best to address the questions raised by the reviewer. Unfortunately, our response could not satisfy the reviewer. We appreciate all the valuable comments from the reviewer, and we will continue optimizing our experiments as well as theoretical modeling of current-induced effects in AFMs. We hope we can have a more comprehensive understanding in the future.

2. *Although the authors mention in their response letter that they were unable to image AFM domains with the quadratic MOKE, they do, on the other hand identify in their PRB 103, L100405 (2021) small AFM domains in comparable Pt-coated α -Fe₂O₃ films of the same nominal thickness using XMLD scans. In this previous work, they even show (nonvolatile) current-induced switching of AFM domains with probably the same current-induced effective fields as they use here at lower current densities to produce the small Néel vector deviations responsible for the higher harmonic signals. Therefore, it would have been indeed more convincing for me if the higher harmonics measurements had been applied as well to the identification and analysis of permanently switched states. Because in this case, this method would have been applied to an undisturbed AFM system that is not affected by a very strong magnetic field of up to 14 T, which generates an additional net magnetization in the AFM state and possibly magnetizes the strongly magn-field susceptible platinum layer.*

Response: We agree that imaging techniques are important for understanding current-induced dynamics in AFMs. However, as we mentioned in the last round of response letter, imaging techniques are not compatible with the harmonic measurements because they work in distinct regimes of AFM dynamics: switching of AFM domains vs. small perturbation of AFM spins. Different from the current-induced switching as in our previous work PRB 103, L100405 (2021), the ac current applied in harmonic measurements only drives the AFM Néel vector slightly deviating from its equilibrium orientation.

3. *This leads me to my last point, which I actually mentioned already in my very first report. The sentence in the abstract "The harmonic measurement technique has never been verified in an antiferromagnetic heterostructure" is incorrect, since harmonic measurement techniques have*

in fact already been used in an antiferromagnetic heterostructure, namely Pt/CuMnAs, to identify 180-degree current-induced Néel vector switching (Nature Communications 9, 4686 (2018)). Here, the second harmonic signal is the result of both a small current-induced spin-orbit field capable of deflecting the Néel vector away from its equilibrium position of switched AF states.

In my opinion, it is basically possible to characterize any AFM heterostructure by a second harmonic response in which a staggered current induced effective field is able to deflect the Néel vector from its equilibrium positions and in which these equilibrium positions are not along extrema with respect to the relevant linear magnetotransport response.

Therefore, a second harmonic signal could in principle also be generated by a current-induced effective field corresponding to a damping-like torque in contrary to what is said in this paper.

Response: In *Nature Communications* 9, 4686 (2018), they used Pt/CuMnAs, but all the actions of electrical switching and second harmonic measurement happen within the CuMnAs layer while the Pt only serves as a protection top layer. Also, their second harmonic measurement serves as a detection technique for the 180° AFM spin reversal of AFM spin orientation in CuMnAs, not for spin torques. We agree with the reviewer that the sentence in the abstract is not clear enough. We modified it in the revise version to, “However, harmonic measurement of spin-orbit torques has never been verified in antiferromagnetic heterostructures.”

4. *In summary, I still feel that the present work characterizing the 3rd harmonic response to an antiferromagnetic heterostructure, without having fully clarified the 2nd harmonic response, is neither conclusive nor novel enough to be published in Nature Communications.*

However, since I am repeating previous criticisms, perhaps the opinion of an additional reviewer should make the final decision on whether or not this work should be published in Nature Communications.

Response: We thank Reviewer#2 for the suggestion of an additional reviewer to evaluate our work.

Response to Reviewer #4

1. *The manuscript deals with the harmonic characterization of antiferromagnetic materials to identify an electrical approach to characterize the damping-like torque and thermally-induced magnetoelastic effect contributions considering the third harmonic voltage. The material platform is the Pt/alpha-Fe2O3. The manuscript is well written and the topic is timely. I have look into the calculations of the second harmonics voltage and the current description is correct. It captures the essential physics the authors wish to describe. I also do not think it is necessary to image the antiferromagnetic state and I agree with the response of the authors to the referee #1 point 1. On the other hand, I noticed that the authors do not consider thermal effect on the magnetic parameters, for example the amplitude of field and damping like torque have a temperature dependence linked with the magnetic parameters of the antiferromagnets. This aspect has to be commented and checked.*

In conclusion, I support the acceptance of the publication, however I have also additional minor comments listed below which should be discussed/addressed before publication.

Response: We appreciate Reviewer #4’s support and all the comments and suggestions to help improve the quality of our work. We agree that the thermal effect on magnetic parameters should

be addressed. α -Fe₂O₃ has a very high Néel temperature ~ 950 K. Thus, the magnetic parameters, for example the sublattice magnetization, barely depend on temperature in our measurement temperature range. This has been confirmed in our previous work (See Fig. S2 in the Supplementary Materials of *Phys. Rev. Lett.* 124, 027202 (2020)), where the net magnetization measured in SQUID shows no difference at 300 K and 200 K.

In *J. Phys. D: Appl. Phys.* 51 095001 (2018), the authors did temperature-dependent harmonic measurements from 50 to 300 K in Co/Pt multilayers to extract the SOT effective field. The damping-like torque effective field $H_{DL} \propto \frac{\xi_{SH}}{M_s}$, where M_s is the saturation magnetization of the magnetic layer and ξ_{SH} is the spin Hall efficiency of Pt. The authors found that from 50 to 300 K, $\frac{1}{M_s}$ increases by 9% while H_{DL} and ξ_{SH} increase by almost 30%. Thus, it is believed that the increase of spin Hall efficiency of Pt due to the increase of resistivity plays a dominate role of increase of H_{DL} . On the other hand, although current-induced temperature change is minor (< 5 K) based on our COSMOL simulation, we still need to discuss its influence on our harmonic measurement.

In *Nature Nanotech.* 8, 587 (2013), the authors claim that the current-induced heating softens the FM magnetization. Since in FMs with PMA, $R_{AHE} \propto M_z \approx M_s \left(1 - \frac{H^2}{2B_k^2}\right)$. Here B_k is the effective field of perpendicular uniaxial anisotropy. The current-induced heating reduces B_k and M_s by $B_k(T) = B_{k0}(1 - \alpha_k I^2)$ and $M_s(T) = M_{s0}(1 - \alpha_M I^2)$. Then $\Delta R_{AHE} \propto [\alpha_M M_{s0} \left(1 - \frac{H^2}{2B_k^2}\right) + \alpha_k \frac{H^2}{B_k^2}] I^2 \propto I^2$. Thus, $\Delta V = I \Delta R_{AHE} \propto I^3$, which gives rise to a third harmonic contribution.

Back in our case, $R_{TSMR} \propto n_x n_y$. Given that we also have $n(T) = n_0(1 - \alpha_n I^2)$, $\Delta R_{TSMR} \propto 2\alpha_n n_{0x} n_{0y} I^2$ when ignoring the higher order terms. Then we have $\Delta V = I \Delta R_{TSMR} \propto I^3$ with angular dependence $\sin 2\varphi_H$. This contribution can be merged into $V_{3\omega}^{\Delta R} = \frac{1}{8} \Delta V_{TSMR} \sin 2\varphi_H$. In our manuscript, we attribute the $V_{3\omega}^{\Delta R}$ solely to the change of Pt resistivity due to thermal effect. However, any change of magnetic parameters could also contribute to the $V_{3\omega}^{\Delta R}$ term. As we mentioned previously, the current-induced temperature change is minor. Thus, we believe that $V_{3\omega}^{\Delta R}$ is mainly originated from the current-induced resistivity change of Pt. We have modified our wording in the main text and added a section (section 6) in Supplementary Information to better address this point, as shown below.

In the main text (bottom of page 5):

“ $V_{3\omega}^{\Delta R}$ **mainly** originates from the change of Pt resistivity due to the applied current. In previous reports of electrical switching of AFMs, thermally-induced Pt resistivity change has led to saw-tooth shaped artifact in switching signals. **And there could be a very minor contribution to $V_{3\omega}^{\Delta R}$ due to the heating induced soften of magnetization given the very high Néel temperature of α -Fe₂O₃.**”

In the supplement (middle of page 7 and bottom of page 13):

“Besides, heating induced soften of magnetization could also modify the measured TSMR (See Section (6) for more details). In both cases, the change of the resistance $\Delta R \propto I^2$ contributes to the third harmonic voltage.”

“6) Thermal effect on the magnetic parameters

α -Fe₂O₃ has a very high Néel temperature ~ 950 K. Thus, its magnetic parameters, for example the sublattice magnetization, barely depend on temperature in our measurement temperature range. This has been confirmed in our previous work, where the net magnetization measured in SQUID shows no difference at 300 K and 200 K. For temperature dependence of SOT effective field, it has been demonstrated that in HM/FM systems such as Co/Pt multilayers, the monotonic increase of SOT effective field with temperature is mainly due to the increase of spin Hall efficiency of Pt instead of the decrease of saturation magnetization M_s . On the other hand, although current-induced temperature change is minor (< 5 K) based on our COSMOL simulation, we still need to discuss its influence on our harmonic measurement. Since $R_{\text{TSMR}} \propto n_x n_y$, and the current-induced heating reduces n by $n(T) = n_0(1 - \alpha_n I^2)$,³⁷ $\Delta R_{\text{TSMR}} \propto 2\alpha_n n_{0x} n_{0y} I^2$ when ignoring higher order terms. Then we have $\Delta V = I \Delta R_{\text{TSMR}} \propto I^3$ with angular dependence $\sin 2\phi_H$. This contribution can be merged into $V_{3\omega}^{\Delta R} = \frac{1}{8} \Delta V_{\text{TSMR}} \sin 2\phi_H$. However, as mentioned above, the current induced temperature change is minor. Thus, we believe $V_{3\omega}^{\Delta R}$ is mainly originated from the current induced resistivity change of Pt.”

2. *I have the following comments on the manuscript. In the introduction several important work on antiferromagnets are not considered. First, the switching has been demonstrated in heavy-metal (HM)/AFM-metallic bilayers, and in those systems it has been also shown the scalability of the switching process in confined geometry and its potential compatibility with CMOS technology [see for example Shei et al Nature Electronics 3, 92–98 (2020)].*

Response: We thank Reviewer#4 for the comment and have added the Nature Electronics paper as reference #7 in our revised manuscript.

3. *From what I found in literature, the debate on the mechanism of the Néel order switching, which could be induced by spin-orbit torque (SOT) or the magnetoelastic effect is important when considering AFM isolating systems while for metallic systems the debate is related to the contribution of the HM in the resistance change [Meinert et al, Phys. Rev. Applied 9, 064040, 2018,] and the effect of morphological features [Bodnar, et al, Phys. Rev. B 99, 140409(R), 2019]. The authors should expand more on this motivation.*

Response: Reviewer #4 raises an important point. We have added more related references (including the two mentioned here) and modified our manuscript on top of page 2 to make it clear.

“However, there is ongoing debate on the mechanism of the Néel order switching, which could be induced by spin-orbit torque (SOT) or the magnetoelastic effect as well as the artifact signal from heavy metal and the relation to AFM grain morphology.”

4. *May the authors add references to support the following sentence: “Whether harmonic measurement can be used in characterizing the current induced effect in other AFMs is still an open question.”*

Response: As suggested by the review, we have added more references (refs. 22-26) in the middle of page 2 to support our point.

5. Page 4, to fix Fig. d in Fig. 1d.

Response: Thank you. We have fixed that typo.

6. *The authors cite the Supplementary Materials in the following sentence “which is the same as the transverse spin Hall magnetoresistance (TSMR) in DC measurements (see Supplementary Materials for more details)”, I’m not sure to which part they are referring.*

Response: Here, we refer to Eq. S1-3 in Supplementary Information, $V_{1\omega} = I_0 R_0 + \frac{3}{8} I_0^3 \frac{d^2 R}{dI^2} \Big|_{I=0} \approx I_0 R_0$. Thus, the first harmonic measurement is equivalent to the DC measurement. We thank the reviewer for pointing this out. We have modified our main text near the bottom of page 3 to make it clear.

“...in DC measurements (see Eq. S1 in Supplementary Materials for more details)”

7. *When discussing about “the effective anisotropic field of the magnetoelastic effect is induced by thermoelastic stress $\Delta\sigma$ ” the authors should cite O Gomonay and D Bossini *J. Phys. D: Appl. Phys.* 54, 374004, 2021 where a complete theory has been developed in this context.*

Response: We thank Reviewer #4 for this suggestion. We have added this reference (Ref. 40) in our revised manuscript.

Reviewers' Comments:

Reviewer #4:

Remarks to the Author:

The authors have addressed my previous criticisms and I can recommend the manuscript for publication.

Fifth Response to Reviews on Manuscript NCOMMS-21-23560D

Response to Reviewer #4

- 1. The authors have addressed my previous criticisms and I can recommend the manuscript for publication.*

Response: We thank Reviewer #4's support for our work.